# An item sorting heuristic to derive equivalent parallel test versions from multivariate items

**Nicole Göbel**[1,2]*, **Dario Cazzoli**[3,4,5], **Clemens Gutbrod**[2], **René M. Müri**[1,2,4], **Aleksandra K. Eberhard-Moscicka**[1,2,5]*

**1** Perception and Eye Movement Laboratory, Departments of Neurology and BioMedical Research, Inselspital, Bern University Hospital, University of Bern, Bern, Switzerland, **2** Department of Neurology, Inselspital, Bern University Hospital, Bern, Switzerland, **3** Neurocenter, Luzerner Kantonsspital, Lucerne, Switzerland, **4** Gerontechnology and Rehabilitation Group, ARTORG Center, University of Bern, Bern, Switzerland, **5** Department of Psychology, University of Bern, Bern, Switzerland

* aleksandra.eberhard@unibe.ch (AKEM); nicole.goebel@usb.ch (NG)

**Data Availability Statement:** The data and materials for all analyses are available at URL: https://osf/3a4c5/ with DOI 10.17605/OSF.IO/3A4C5.

## Abstract

Parallel test versions require a comparable degree of difficulty and must capture the same characteristics using different items. This can become challenging when dealing with multivariate items, which are for example very common in language or image data. Here, we propose a heuristic to identify and select similar multivariate items for the generation of equivalent parallel test versions. This heuristic includes: 1. inspection of correlations between variables; 2. identification of outlying items; 3. application of a dimension-reduction method, such as for example principal component analysis (PCA); 4. generation of a biplot, in case of PCA of the first two principal components (PC), and grouping the displayed items; 5. assigning of the items to parallel test versions; and 6. checking the resulting test versions for multivariate equivalence, parallelism, reliability, and internal consistency. To illustrate the proposed heuristic, we applied it exemplarily on the items of a picture naming task. From a pool of 116 items, four parallel test versions were derived, each containing 20 items. We found that our heuristic can help to generate parallel test versions that meet requirements of the classical test theory, while simultaneously taking several variables into account.

## Introduction

Parallel versions of psychometric tests are typically used to control for systematic measurement errors, such as biases due to learning effects or fatigue. They are often used in longitudinal studies, for example to assess the course of neurological diseases. To ensure that parallel test versions are equivalent, they must capture the same characteristics while, at the same time, using different items and showing a comparable degree of difficulty.

Thereby, classical Test Theory (CTT) assumes that each item is equally difficult [1]. While CTT attempts to estimate the true score of the characteristic to be measured based on the responses in several items and focuses on the accuracy of a given measurement as well as on the magnitude of the measurement error, it can only address one variable at a time [2]. At the same time, the selection of the test items is typically performed iteratively by hand. Though,

**Funding:** "The study was supportedby the Swiss National Science Foundation Grant no: 175615. There was no additional external funding received for this study."

**Competing interests:** The authors have declared that no competing interests exist.

Gulliksen proposed already in 1950 to plot item analysis results, such as the reliability and validity indices of the item scores, in order to select the best items [3].

Unlike CTT, item response theory (IRT) does not assume that each item is equally difficult [1]. The difficulty of each item is treated as an information that is incorporated into the item characteristic curves (ICC). Thereby, the ICC represents the difficulty of the item by the probability curve of answering the item correctly as a function of the subject's underlying trait. As opposed to CTT, which focuses on the test, IRT focuses on the item. To date, several automated algorithms for constructing parallel test forms that make use of the item information function from IRT have been suggested [4–9]. IRT assumes manifest variables (i.e., the response behavior to test items) and a latent variable (i.e., an underlying characteristic of the subjects). Despite its clear advantages (e.g., items with different difficulty levels and sample independence of test characteristics), IRT approaches usually assume only one latent variable, which is reflected in the correlation between the manifest variables [10–13].

Importantly, both CTT and IRT usually consider only one variable, respectively one latent variable at a time. In practice, however, it can be of interest to describe the items with respect to several variables, to derive more than one latent variable, and to divide these items among parallel test versions such that they are comparable with respect to all variables. In this regard, it might be of interest to consider approaches dealing with the detection of multivariate similarity of items that are applied in other domains. For example, in order to select stimuli for experiments with a factorial design satisfying an extensive list of experimental requirements, Guasch et al. [14] proposed a method that is halfway between a manual and an automated stimuli selection. In this approach, small and tight clusters of words matching within the variables of interest were identified by means of k-means clustering.

To date, several automated methods for multivariate stimulus selection were proposed [15–18]. For example, in the optimization approach, items selected for previous test versions are removed from the item pool, thus causing the later test versions to be less likely parallel than the earlier ones. Chen et al. argue that, in contrast, a random search approach may result in more uniform test versions [7]. Yet, Guasch et al. [14] highlighted the advantages of his half-automated procedure over the automated methods: "Picking items by hand is tedious (. . .)" and "a fully automated selection (. . .) leaves open the question if a better solution could have been found".

This approach is not only relevant for stimulus selection, but it could also be of advantage for multivariate item selection in the context of parallel test generation. In their work, Guasch et al. [14] used k-means clustering, yet other dimension-reduction methods are also applicable, e.g., Multidimensional Scaling (MDS) in the case of mixed data [19], or newer methods for numerical data such as e.g., Stochastic Neighbor Embedding (t-SNE) [20]. Given its wide and common use in psychological research, in the current work we used Principal Component Analysis (PCA) as a dimension-reduction method. The main advantages of PCA are its suitability for numerical data, its lack of requirements with respect to distributional assumptions, its suitability for highly correlated variables, along with its usefulness even in the case of relatively large number of variables with respect to observations [21–23].

In the following, we present the utility of PCA as a dimension-reduction method for selecting multivariate items for parallel test versions. To this end, a heuristic is first presented as a general procedure and further applied on a practical example. Our example deals with the development of four parallel test versions of the Bern word-finding test (B-WFT, unpublished), a picture naming test used to detect word-finding disorders. Up to date, the B-WFT consisted of two test versions. By using its original pool of 116 multivariate items, and by applying principal component analysis (PCA) as a dimension-reduction method, we created a

set of four parallel test versions (ABCD). Finally, these test versions were tested for multivariate equivalence, parallelism, reliability, and internal consistency.

## Method

The studies involving human participants were reviewed and approved by the Ethics Committees of the Cantons of Luzern and Bern, Switzerland (EKNZ 2015–256; KEK BE 151/15). The participants provided their written informed consent to participate. The data were analyzed anonymously for this study.

To generate parallel test versions, a sufficient pool of items is required, in order to allow for the exclusion of potential outliers. Moreover, when using PCA within the proposed heuristic, the items must be described by several, numeric, intercorrelated variables. These are often obtained by presenting the items to a number of subjects, asking them to name and rate the items on several dimensions. The general procedure requires two types of data tables. One table contains the raw data of each subject, and it is primarily used to check the results. Another table contains all items with the respective mean values of the variables, and it is mainly required for dimension-reduction. If these prerequisites are met, the following heuristic can be applied:

1. Inspect correlations between items: Compare pairwise all numeric variables of the test items in order to ascertain whether there are any correlations between them. If there are no correlatons between the items, the heuristic may not work.

2. Identify outliers: Since dimension-reduction methods can be sensitive to outliers, it is advisable to identify them and consider their exclusion from the item pool. Outliers can be identified by means of robust PCA [24], a modification of the classical PCA [25].

3. Reduce dimensions: A common approach to reduce dimensions in binary or numerical data is PCA. It reduces the dimensionality of the multivariate data set, while accounting for as much of the original variance as possible. This results in a new set of variables, so-called principal components (PC), which are linear combinations of the original variables. PC1 represents the direction of the data cloud with the highest variance, while PC2 represents the direction of the data cloud with the second highest variance [21].

4. Plot and group similar items: By plotting the resulting 2D map of the first two PCs, similar items can be identified by their spatial proximity, hence can be easily grouped together. Items with medium item difficulty usually have the best discriminatory power [26]. Therefore, it is recommended to select the items from the center of the resulting plot and ignore the items in the border areas. Group size depends on how many parallel test versions are to be derived. Additional items in a group allow for a better fine-tuning. If, for instance, four parallel test versions with 10 items each would be required, one would have to identify 10 groups with at least four items plotted close to each other.

5. Assign items to test versions: From every group, select exactly one item per parallel test version. Try to achieve additional balance through quasi-randomization (e.g., use assigning order ABCD, BCDA, CDAB, DABC). Check for equality of mean values and variances by generating correlation plots or boxplots. Use extra items to replace less suitable ones.

6. Check resulting test versions: To assess whether all variables were equally well addressed by the PCA, perform a multivariate comparison of the final parallel test versions, such as analysis of variance (MANOVA) or multivariate Kruskal-Wallis. If any of the variables becomes significant, test versions would be different with respect to individual variables.

Also, perform the customary analyses to determine whether the requirements of parallelism, reliability, and internal consistency have been met.

## Results

To illustrate this item sorting heuristic, a practical example is provided. All statistical analyses were performed with R, Version 4.2.2. Please ensure that you download all files, update the path to the data set in the R file, and load the R packages, which are required for the libraries used. The data and materials for all analyses are available at URL: https://osf.io/3a4c5 with DOI 10.17605/OSF.IO/3A4C5.

### 1. Inspect correlations of items

In this example, four parallel test versions of the B-WFT, with 20 items each, were to be derived from an already existing pool of items. The original item pool consisted of 116 simple, black-and-white line drawings. Half of the drawings represented living and the other half non-living objects. As an additional requirement, parallel test versions had to contain 10 living and 10 non-living items. Items were selected from the image corpus by Snodgrass and Vanderwart [27], the image archive of Dr. Dorothea Weniger, and a pool of line drawings by the Inselspital, Bern University Hospital (Berger EM et al. [Unpublished], S1 File).

The drawings were presented to 52 healthy, (Swiss-)German speaking adults ($M$ = 42.79, $SD$ = 21.17, age range 22–81 years, 27 male and 25 female, 3 left-handers), with an average of 14.12 years of education ($SD$ = 2.55). Since the procedure was applied to already existing data, no influence on the sample size could be taken. The subjects were asked to name all of the 116 items.

Variables measured are presented in Table 1: object class, naming agreement, image agreement, image complexity, object familiarity, accuracy, and response time. The overall Kaiser-Meyer-Olkin value was sufficient to perform factor analytic procedures ($KMO$ = 0.608). As indicated in Table 2, response time was strongly and positively correlated with accuracy ($r$ = 0.68), and accuracy was strongly and positively correlated with image agreement ($r$ = 0.71). Hence, the requirement of inter-variable correlation for PCA was met.

**Table 1. Variables collected.**

| Variable | Abbreviation | Definition |
|---|---|---|
| Object class | ObCl | Whether the item is living (e.g., plant) or non-living (e.g., tool). |
| Naming agreement | NaAg | The number of different terms used to name an item. The higher the value, the more often the item was named differently. |
| Image agreement | ImAg | How well the image matched with the subject's idea of the depicted concept. This was rated by the subjects on a Likert scale ranging from 1 to 5, with lower values corresponding to a higher ImAg. |
| Image complexity | ImCo | The level of detail in which the item was drawn. This was rated by the subjects on a Likert scale ranging from 1 to 5, with lower values indicating fewer details and strokes. |
| Object familiarity | ObFa | Rated by the subjects on a Likert scale ranging from 1 to 5. The lower the value, the more familiar the subjects were with a given item. |
| Accuracy | ACC | This variable was computed as an index, i.e., the proportion of subjects who identified and named a given item correctly. A higher value indicates a higher proportion of correctly named items. |
| Response time | RT | How long subjects took to answer, in milliseconds. |

Note. Except for object class, all variables consisted of mean values based on 52 healthy subjects.

**Table 2. Pairwise comparisons between all variables.**

|  | ObCl | NaAg | ImAg | ImCo | ObFa | ACC | RT |
|---|---|---|---|---|---|---|---|
| ObCl | – |  |  |  |  |  |  |
| NaAg | -.26 ** | – |  |  |  |  |  |
| ImAg | .22 * | .04 | – |  |  |  |  |
| ImCo | .07 | .10 | .01 | – |  |  |  |
| ObFa | .41 *** | -.15 | .17 | .54 *** | – |  |  |
| ACC | .18 | .00 | .71 *** | .07 | .27 ** | – |  |
| RT | -.06 | .36 *** | .59 *** | .12 | .08 | .68 *** | – |

Note. Correlations indicate that PCA is applicable. ObCl = object class, NaAg = naming agreement, ImAg = image agreement, ImCo = image complexity,

ACC = accuracy, RT = response time. Significance levels:

*** <0.001,

** <0.01,

* <0.05.

Test statistics based on Pearson's product moment correlations and polychoric correlation coefficients.

## 2. Identify outliers

Classical PCA tilts the PCA subspace towards outliers. Hence, to identify potential outliers in advance, robust PCA was applied, which first fits most of the data and subsequently flags data points that deviate from the main body of data (Fig 1). Regular observations have both a small orthogonal distance (y-axis) and a small score distance (x-axis). Items with a high score distance but a small orthogonal distance (leverage points) can improve the accuracy of the fitted

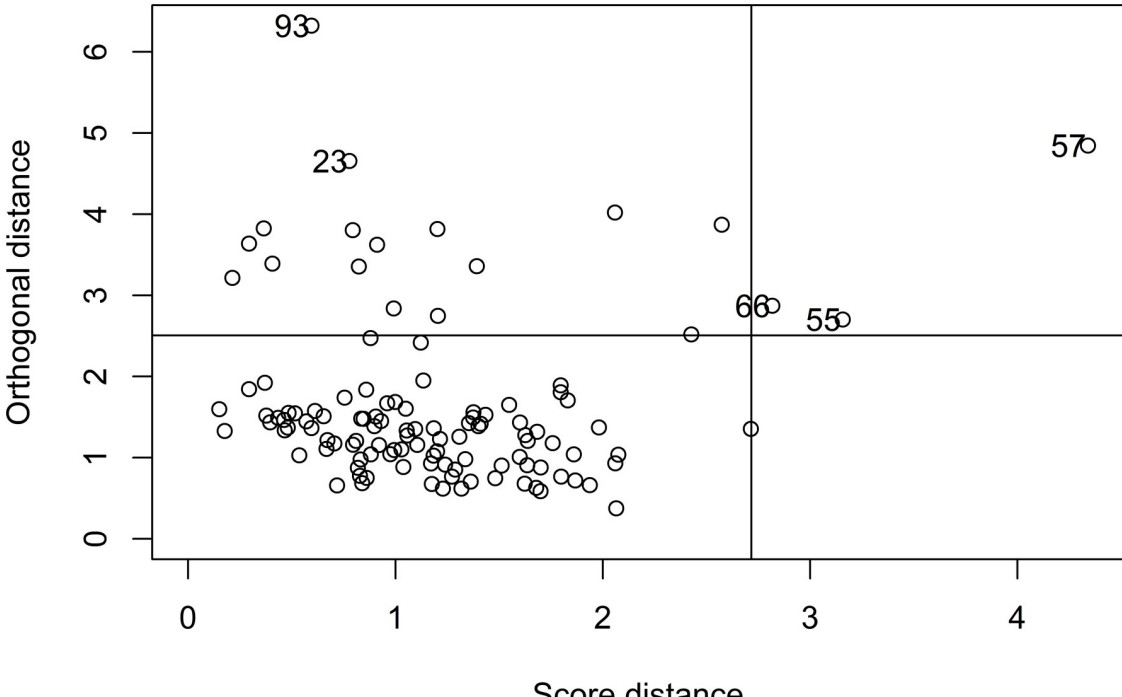

**Fig 1. Plot of robust PCA.** Regular observations have both a small orthogonal distance (y-axis) and a small score distance (x-axis). Items 93, 23, 66, 55, and 57 can be considered as outliers.

PCA subspace. In our example, items 93 and 23 are orthogonal outliers with a large orthogonal distance but a small score distance. Items 66, 55, and 57 have both a large orthogonal distance and a large score distance (bad leverage points) [28]. Simply put, items 93, 23, 66, 55, and 57 can be considered as outliers, which may potentially be excluded from the further procedure. In the current example, due to the limited number of items, we retained all items in the item pool.

### 3. Reduce dimensions of items

For dimension-reduction, classical PCA with data scaling was chosen. A total of six PCs were identified, of which the first three PCs accounted for 84.16% of the variance in the data. PC1 explained most of the variance (40.79%), and included the correlating variables response time, accuracy, and image agreement. PC2 explained the second largest part of the variance (24.57%) and contained the correlating variables image complexity and object familiarity. PC3 explained the third largest part of the variance (18.79%) and contained the variable naming agreement.

### 4. Plot and group similar items

A 2D map of the data was obtained by plotting the first two PCs, i.e., PC1 and PC2 (Fig 2). The items that are similar with respect to the variables lie closely together in the graph. In the current example, the aim was to generate four parallel test versions with 20 items each. Therefore, 20 groups of at least four similar items were identified. Note that grouping is performed visually and may therefore slightly vary for subjective reasons. However, even though item selection may be an iterative process, the quality of the resulting test versions is objectively evaluated in the final step.

The embedded biplot (arrows in Fig 2) illustrates how the variables contributed to the distribution of the items in the 2D plot. Numbers from 1 to 116 represent the items. Arrows represent the direction in which the corresponding variables run. Small-angled arrows (such as, e.g., for response time, image agreement, and accuracy) indicate high correlations between the corresponding variables. Items that follow the direction of a given arrow have high values in the corresponding variable (e.g., item 110 has high values in image complexity and object familiarity).

PC1 and PC2 are represented by the axes. Since PC3 consists only of naming agreement, its direction can be read from the corresponding variable arrow. The fact that naming agreement has a shorter arrow indicates that this variable was not as well captured by the 2D plot as the other variables. Nevertheless, the arrow corresponding to naming agreement gives an overall idea of the direction of PC3.

### 5. Assign items to test versions

In the next step, the items were distributed into four test versions (i.e., A, B, C, and D). The distribution procedure followed a quasi-randomized assignment order (i.e., ABCD, BCDA, CDAB, DABC), considering that each version had to contain 10 living and 10 non-living objects.

Boxplots of the test versions (Fig 3) indicate that test version B may differ from the other versions A, C, and D with respect to the variable response time. Test version C shows less variance in response time and more variance in accuracy than the other test versions. Also, accuracy shows a strong ceiling effect, indicating that the task was too simple for the 52 healthy subjects. The ceiling effect is not surprising, since the test was aimed at identifying word finding disorders in patients. Also, other variables (e.g., response time) show a slightly skewed

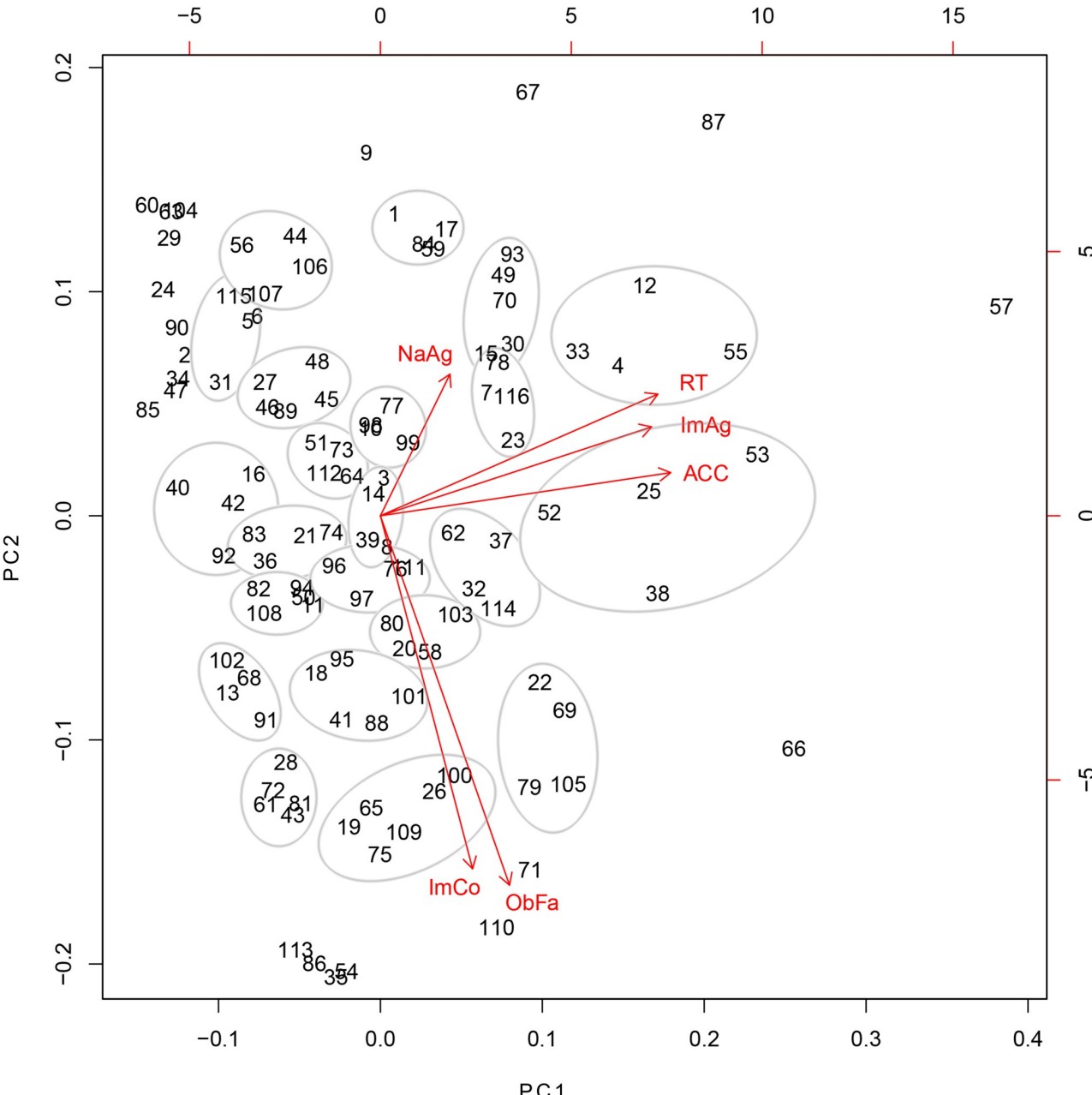

**Fig 2. Biplot of PC1 and PC2.** Axes: left and bottom = normalized principal component scores of each item; top and right = factor loadings of each variable. Items are symbolized by numbers ranging from 1 to 116. Spatial proximity of the items corresponds to their similarity with respect to the considered variables. 20 groups of at least four similar items are circled in grey. Arrows = direction of variables; abbreviations on top of the arrows = variable names (NaAg = naming agreement, RT = response time, ImAg = Image Agreement, ACC = accuracy, ImCo = image complexity, ObFa = object familarity); short arrows = variables that are not well captured by the 2D plane; right-angled arrows = no correlation between the corresponding variables; small-angled arrows = high correlations between the corresponding variables.

distribution, suggesting the necessity of performing a transformation. Although according to the visual inspection of the quantile-quantile plot, the log- as well as the box-cox-transformed variable response time seemed normally distributed, the normal distribution assumption had to be rejected with the Kolmogorov-Smirnov (K-S) test ($p$-value $<.001$).

## Image Complexitiy

## Image Agreement

## Naming Agreement

## Response Time

## Accuracy

**Fig 3. Boxplots of the variables of concern.** The y-axis corresponds to units and the x-axis to test versions A, B, C, D, and unused items X that were not assigned to any test version.

If differences in mean values or in variances between the four test versions ABCD would have become visible in the boxplots for any of the variables, the causing items could have been exchanged between versions.

## 6. Check resulting tests versions

There are different criteria that need to be met for test versions to be considered as equivalent. Parallelism is given if mean values and variances are equivalent [29]. Parallelism can be tested with the Bradley-Blackwood test, and equality of means with the two-one-sided tests (TOST). ANOVA can provide insight as to whether the subjects differ in their performance on the parallel test versions, and on how the residuals behave. If test versions are parallel, Pearson's product moment correlations can be taken as an estimate of the reliability of the test versions. Furthermore, Cronbach's alpha provides information about internal consistency. Since the proposed method to generate parallel tests takes multiple variables into account, we also propose a parametric multivariate analysis of variance (MANOVA) or a non-parametric multivariate Kruskal-Wallis test as a mean to compare test versions.

**Multivariate equivalence of the items.** For a multivariate comparison of the test versions ABCD, both a MANOVA and a multivariate Kruskal-Wallis (MKW) test were performed on the items' data, since the normality assumption was rejected by the K-S test. The parallel test version was defined as a treatment factor, and the following as response variables: object class,

**Table 3. Multivariate comparison of the parallel test versions ABCD.**

|  | ObCl | NaAg | ImAg | ImCo | ObFa | ACC | RT | All |
|---|---|---|---|---|---|---|---|---|
| MANOVA | 1.00 | 0.75 | 0.99 | 0.84 | 0.80 | 0.95 | 0.99 | 1.00 |
| MKW | 1.00 | 0.54 | 0.99 | 0.97 | 0.87 | 0.86 | 0.95 | 1.00 |

Note. Abbreviations: ObCl = object class, NaAg = naming agreement, ImAg = image agreement, ImCo = image complexity, ACC = accuracy, RT = response time, MKW = Multivariate Kruskal-Wallis test. Numbers are indicating *p*-values.

naming agreement, image agreement, image complexity, object familiarity, accuracy, and response time. Mean values of the response variables were based on the results of the 52 subjects described earlier. If a given response variable had been significant in the MANOVA or MKW test, it would have meant that at least one test version was different from the others concerning this particular variable. Statistical power was calculated post-hoc with G*Power, version 3.1.9.2, to be 0.96. Parameters considered to assess statistical power were effect size of 0.1, total sample size of 116 items, 4 test versions, and 7 response variables.

As shown in Table 3, the test versions did not significantly differ on any of the considered variables ($p = 1.00$ for ABCD for all variables), with naming agreement showing the lowest *p*-value (MANOVA: $p = 0.75$, MKW: $p = 0.54$). Naming agreement constituted PC3 and was therefore not particularly well represented by the plot of PC1 and PC2. Yet, it is important to note that the *p*-value is not the probability that the null hypothesis is true, i.e., that the parallel tests are equal [30]. For this reason, it is indispensable to examine the confidence intervals; in this case, however, this would lead to six pairwise comparisons for each of the seven variables. For instance, the confidence interval for the difference in response time between version B and C ranged from -253 to 302 milliseconds. This illustrates that the null hypothesis cannot be rejected, but also that the null hypothesis is not necessarily correct. Rather, the question must be asked as whether a difference of 300 milliseconds would be acceptable to be considered as similar enough. Data of a larger number of subjects would help to reduce the confidence interval.

**Bradley-Blackwood test for parallelism.** From now on, we will focus on the raw data of all subjects in response time. This variable was chosen for practical reasons. To control all variables would go beyond the scope of this paper. Accuracy is not considered suitable as the scores obtained by healthy subjects indicated a ceiling effect. The ceiling effect is not surprising since the test used in the current example is aimed at identifying word finding disorders in patients.

As stated in the section "Multivariate equivalence of the items", parallelism cannot be automatically assumed if the null hypotheses of equality of the observed means cannot be rejected. The Bradley-Blackwood paired-samples omnibus test [31], as proposed by García-Pérez [29], is nevertheless valuable in order to indicate possible differences in means and also variances between test versions. In the present example, this test is being applied with the R-function from the textbook by Hedderich & Sachs [32].

Since the regression model implied in this test assumes a normal distribution of the data, the quantile-quantile plot was consulted first (Fig 4). The output of the plot confirmed the necessity of performing a transformation of the response time before further analyses could be performed. Otherwise, the assumption of normal distribution could not have been warranted. We found logarithmic and box-cox-transformation to deliver a very similar result.

The Bradley-Blackwood test indicated no significant differences in response time between the mean values and variances of the ABCD parallel test versions. Nonetheless, it must be

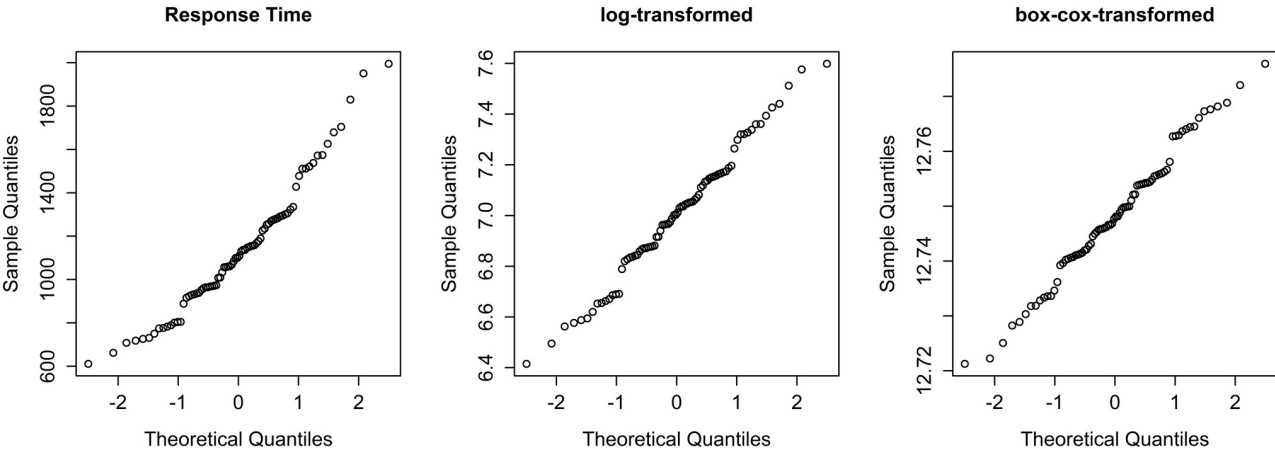

**Fig 4. Quantile-Quantile plot.** Response time (left), logistic-transformed response time (middle), and box-cox-transformed response time (right).

noted that versions B and C are approaching a weak trend of being significantly different ($p = 0.16$, Table 4).

**Two one-sided tests (TOST) for equivalence of means.** In order to investigate the equality of the test versions in more detail, the two one-sided test (TOST) procedure was applied as described in the tutorial by Lakens, Scheel and Isager [33]. In addition to testing against zero, TOST can be used to check for equivalence and to reject the presence of a smallest effect size of interest (SESOI). The TOST procedure helps to ascertain whether an observed effect is surprisingly small, considering that there is a real effect at least as extreme as the SESOI. The TOST procedure requires the following input: means, and standard deviations of the compared test versions, sample size, lower and upper equivalence bounds expressed as standardized mean differences (Cohen's d), alpha level (default = 0.05), and whether the equality of variances assumption is expected to be met.

With lower equivalence bounds of -0.6 and upper equivalence bounds of 0.6, all equivalence tests were significant in our parallel test versions. This means that the test versions do not differ by more than 60% of a standard deviation concerning response time. According to Cohen [34], this would represent a medium effect size. The four test versions can therefore be considered as parallel.

**ANOVA model and analysis of residuals.** To evaluate whether the subjects differed in their performance on the parallel test versions, and to analyze how the residuals behave, the

**Table 4. Bradley-Blackwood paired-samples omnibus test on the log-transformed response times of the parallel test versions ABCD.**

| versions | mean | variance | F | *p*-value |
|---|---|---|---|---|
| A—B | -0.032 | 0.0337 | 1.143 | .33 |
| A—C | 0.016 | 0.0212 | 0.366 | .70 |
| A—D | -0.028 | 0.0195 | 1.341 | .27 |
| B—C | 0.048 | 0.0342 | 1.893 | .16 |
| B—D | 0.004 | 0.0294 | 0.041 | .96 |
| C—D | -0.044 | 0.0299 | 1.713 | .19 |

Note. *p*-values refer to differences between both mean values and variances.

**Table 5. Pairwise comparisons of all parallel test versions.**

|   | A | B | C | D |
|---|---|---|---|---|
| **A** | – | | | |
| **B** | .70 | – | | |
| **C** | .80 | .71 | – | |
| **D** | .82 | .76 | .74 | – |

Note. All correlations are significant at $p$ = .001. Test statistics are based on Pearson's product moment correlations.

data were analyzed using the following mixed effects model:

$$Y_{ij} = \mu + \alpha_i + \beta_j + \gamma_k + \epsilon_{ijk}. \tag{1}$$

$Y_{ij}$ are the logarithmically transformed response times, $\alpha_i$ is the fixed effect of parallel test versions, $\beta_j$ is the random effect of subjects, $\gamma_k$ is the fixed effect of age group. Model assumptions are: $N(0, \sigma_\beta^2)$ and $N(0, \sigma^2)$. Here, subject is a random block factor, since a random sample of subjects were tested on all items contributing to the parallel test versions.

The ANOVA output of the ABCD test versions indicated highly significant differences between the subjects ($p$ <0.001, $CI$ = 0.160, 0.244), almost significant differences between the age groups ($p$ = 0.052, $CI$ = 0.002, 0.235), and no significant differences between the parallel test versions ($p$ = 0.120, $CI_{A-B}$ = -0.013, 0.078, $CI_{A-C}$ = -0.061, 0.029, $CI_{A-D}$ = -0.017, 0.073).

The analysis of the residuals indicated that the error variances were equally distributed, and that the errors, as well as the random effect of subjects, were approximately normally distributed.

**Pearson's product moment correlations of test versions.**  Correlations between the parallel test versions ranged from 0.70 to 0.82 (Table 5). Since the ABCD test versions have been found to be parallel, the correlation between test versions can be held for an estimate of the reliability of the test [29].

**Cronbach's alpha for internal consistency.**  According to Cronbach [35], *alpha* is the mean of all possible split-half coefficients. Hence, it is the value expected when two random samples of items from a given pool (like, e.g., those in the given example of parallel test versions) are correlated. As such, Cronbach's alpha is an indicator of homogeneity within parallel test versions [35].

Cronbach's alpha was calculated for the response times of the 52 subjects for each of the four parallel test versions. The resulting internal consistencies of the parallel test versions ABCD were good, with $\alpha$ ranging between 0.81 and 0.84 (A: 0.81, B: 0.81, C: 0.81, D: 0.84).

## Discussion

We devised and tested a heuristic to generate multiple, equivalent parallel test versions from a multivariate pool of items. The core of our heuristic is based on the reduction of multivariate items to two dimensions, which is neither a CTT nor an IRT approach, but could be considered related to IRT. Similar to IRT, we assume that items vary in difficulty, which in our case is represented by the different values of the items on the first principal component (Fig 2).

Our items and variables, on which these principal components are based, were generated, and collected in the past. Among them, Response Time and Accuracy would be the variables most comparable to typical item response variables. These two, along with Image Agreement, form the first principal component, which accounts for 40.79% of the variance in the

data. This noteworthy relationship might have escaped our attention, had we used the IRT approach.

Given that unidimensionality is considered a prerequisite for IRT and bidimensionality is a prerequisite for our dimension-reduction heuristic, it seems a legitimate question how high the proportion of explained variance should be in order to be considered an indicator of unidimensionality or bidimensionality.

Hattie [36] refers to authors who propagate 20% or 40% for unidimensionality. However, in his conclusion, he suggests that it may be unrealistic to search for indications of unidimensionality, and that the test score is basically a weighted composite of all the underlying variables. We share and address this idea by suggesting the use of multivariate items and dimension-reduction procedures.

A common criterion for dimension-reduction methods is to retain as many components until about 70–90% of the variance is explained [21–23]. In our heuristic, by relaying on two principal components, we achieve an explanation of the variance of 65.36%. Whereas the first principal component (composed of Response Time, Accuracy, and Image Agreement) accounts for 40.79% of the variance in the data, the second principal component (composed of Image Complexity and Object Familiarity) accounts for 24.57% of the variance in the data. Even if the latter has no direct influence on Response Time or Accuracy, it may still influence the subjects' responses, e.g., in the form of faster fatigue over the duration of a test.

In summary, while IRT usually assumes one latent variable, we assume two principal components from a variety of variables that ideally cover a large portion of the variance in the data. This approach helps to represent items on a two-dimensional graph, in which similarity of items is represented by spatial proximity.

For the selection process, we proposed a semi-automatic approach as opposed to fully automatic methods. While in the latter, similar items such as e.g., "hen" and "rooster" might be potentially selected in the same parallel test version (shall semantic similarity not be expressed as a distinct variable), a semi-automatic approach allows correction of such biases.

Within the proposed heuristic, PCA appears to be a good choice as a dimension-reduction method. Based on the plots of the first two PCs, similar items could easily be identified and further assigned to parallel test versions. An important advantage of PCA is that it does not require any distribution assumptions. However, since this approach is based on the empirical covariance matrix of the data, it is sensitive to outliers. Given that in the current example we were restricted by a limited number of items, we retained the outliers in the item pool.

In order to confirm that the resulting test versions met multivariate as well as CTT criteria (i.e., multivariate equivalence, parallelism, equivalence of means, reliability, and internal consistency), we performed both multivariate procedures (i.e., MANOVA and multivariate Kruskal-Wallis) as well as univariate procedures (i.e., Bradley-Blackwood test, TOST, ANOVA, Pearson's product moment correlations, and Cronbach's alpha), most of which are known from CTT. The usual iterative process of the item selection of the CTT could be omitted. With the example provided, we could demonstrate that the proposed heuristic generated a set of four parallel test versions ABCD, satisfying requirements of multivariate equivalence, parallelism, reliability, and internal consistency.

Applying a MANOVA as well as a multivariate Kruskal-Wallis test, which better fits our data, we simultaneously examined all variables, and demonstrated that the ABCD test versions did not significantly differ with respect to any of the variables considered. As previously noted, the *p*-value does not represent the probability that the parallel tests are equal. We therefore propose to examine confidence intervals. In the present example, for instance, the confidence interval for the difference in response time between version B and C of the ABCD parallel test versions ranges from -253 to 302 milliseconds. Even though the null hypothesis (i.e., that the

test versions are identical) cannot be rejected, response times in the respective test versions can still differ up to 302 milliseconds. Hence, the critical question to be asked in parallel test generation may be as to which difference is still acceptable to be considered as similar enough. Data from more subjects would clearly help reducing the confidence interval. However, our item set as well as the parallel test versions have an exemplary character here, hence are solely used for demonstration purposes. Further application in other data sets needing parallel version creation would show other peculiarities, but the methodology seems flexible enough to adapt to these.

The distinctive feature of the proposed heuristic is that it allows for the generation of multiple test versions while taking several variables into account that have more than one underlying latent variable. Yet, for reasons of practicability, we did not assess all variables with all methods. In our verification approach, except for MANOVA and multivariate Kruskal-Wallis, we focused on the variable response time. Nonetheless, each of these methods could be extended to any further variables.

When checking for parallelism, the parallel test versions passed the Bradley-Blackwood test. The ANOVA results indicated that differences in response time between the subjects were highly significant, and differences between the age groups were almost significant; critically, however, differences between the parallel test versions were not significant. Pearson's product moment correlations, as well as Cronbach's alpha, showed good results for the parallel test versions. Given parallelism, these can be considered reliability and internal consistency measures. However, Cronbach's alpha is a typical CTT measure that reflects the degree to which items within a test version are similar with respect to that dimension. This is not central to our heuristic; on the contrary, we want to allow for item difficulty to vary within a test version, while not leading to different difficulty distributions across test versions. The fact that we nevertheless obtained good results in terms of internal consistency is probably due to the homogeneity of the items in terms of response time and to the fact that the items were mostly selected from the center of the biplot.

Regarding the parallel test versions in our example, the next step shall entail a validation with aphasic patients, with a focus on the variable accuracy. Simulation studies and replications using different kinds of data and dimension-reduction methods, such as Multidimensional Scaling (MDS) in the case of mixed data [19], or Stochastic Neighbor Embedding (t-SNE) [20] in the case of numerical data, would further help to prove the generalizability of the proposed heuristic.

Using an example data set, we demonstrated that PCA can be applied to derive equivalent parallel test versions while accounting for the multivariance of the items.

## Supporting information

**S1 File. Berger EM et al. [Unpublished].** Normierung eines Benenn- und semantischen Entscheidungstests für biologische und manipulierbare Objekte in deutscher Sprache. (PDF)

## Acknowledgments

We would like to thank Dr. Dorothea Weniger for her expert advice in linguistics. We would also like to thank Dr. Lea Jost for valuable discussions on methodology.

## Author Contributions

**Conceptualization:** Nicole Göbel, Clemens Gutbrod, René M. Müri, Aleksandra K. Eberhard-Moscicka.

**Data curation:** Nicole Göbel, Clemens Gutbrod.

**Formal analysis:** Nicole Göbel, Aleksandra K. Eberhard-Moscicka.

**Funding acquisition:** René M. Müri.

**Methodology:** Nicole Göbel, Dario Cazzoli, Aleksandra K. Eberhard-Moscicka.

**Project administration:** René M. Müri, Aleksandra K. Eberhard-Moscicka.

**Resources:** René M. Müri.

**Supervision:** Aleksandra K. Eberhard-Moscicka.

**Validation:** Nicole Göbel, Aleksandra K. Eberhard-Moscicka.

**Visualization:** Nicole Göbel.

**Writing – original draft:** Nicole Göbel, Aleksandra K. Eberhard-Moscicka.

**Writing – review & editing:** Nicole Göbel, Dario Cazzoli, Clemens Gutbrod, René M. Müri, Aleksandra K. Eberhard-Moscicka.

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
