## [Decision Letter · Decision Letter 0]

28 Oct 2022

PONE-D-22-04765An item sorting heuristic to derive equivalent parallel test versions from multivariate itemsPLOS ONE

Dear Dr. Göbel,

Thank you for submitting your manuscript to PLOS ONE. After careful consideration, we feel that it has merit but does not fully meet PLOS ONE’s publication criteria as it currently stands. Therefore, we invite you to submit a revised version of the manuscript that addresses the points raised during the review process.

We look forward to receiving your revised manuscript.

Kind regards,

Alessandro Barbiero, Ph.D. in Statistics

Academic Editor

PLOS ONE

Journal Requirements:

“Nicole Göbel had financial support by the SNF Grant No: 175615.”

“We would like to thank Dr. Dorothea Weniger for her expert advice in 440 linguistics. We would also like to thank Dr. Lea Jost for valuable discussions on 441 methodology. The study was supported by the Swiss National Science Foundation 442 Grant no: 175615”

“Nicole Göbel had financial support by the SNF Grant No: 175615.”

5. We note that you have referenced (Berger E-M.et al. [21]) which has currently not yet been accepted for publication. Please re We note that you have referenced (ie. Bewick et al. [5]) which has currently not yet been accepted for publication. Please remove this from your References and amend this to state in the body of your manuscript: “Berger E-M et al. [Unpublished]”) as detailed online in our guide for authors

http://journals.plos.org/plosone/s/submission-guidelines#loc-reference-style move this from your References and amend this to state in the body of your manuscript: (ie “Bewick et al. [Unpublished]”) as detailed online in our guide for authors http://journals.plos.org/plosone/s/submission-guidelines#loc-reference-style

Reviewers' comments:

Reviewer's Responses to Questions

**Comments to the Author**

1. Is the manuscript technically sound, and do the data support the conclusions?

Reviewer #1: Partly

2. Has the statistical analysis been performed appropriately and rigorously? 

Reviewer #1: Yes

3. Have the authors made all data underlying the findings in their manuscript fully available?

Reviewer #1: Yes

4. Is the manuscript presented in an intelligible fashion and written in standard English?

Reviewer #1: Yes

5. Review Comments to the Author

Reviewer #1: Thank you for the opportunity to serve as a peer reviewer for this interesting submission describing "An item sorting heuristic to derive equivalent parallel test versions from multivariate items". Below I have made recommendations for numerous improvements to be made by the authors to the manuscript before it can merit acceptance for publication in PLoS One journal.

1. In the introduction section you have referenced various existing item response theory (IRT)-based algorithms, that are reported in the extant literature, for constructing parallel test forms. Given the well known advantages of IRT over classical test theory (CTT) methods, how then do you justify a heuristic that is predominantly founded on CTT? Please enumerate the advantages that your CTT-based heuristic is expected to have over the published IRT-based algorithms? The authors need to build a stronger justification for a CTT-based heuristic.

2. Before applying the PCA, why did the authors not begin by deriving the Kaiser–Meyer–Olkin (KMO) test to confirm adequacy of the data as a source of factor-analytic correlation matrices?

3. In what ways is PCA (even robust PCA) superior to other CTT methods such as Horn's parallel analysis or Velicer's minimum average partial (MAP) test, as a foundation for the proposed heuristic?

4. The authors do not always specify which correlation coefficients they utilized in their practical application of the proposed heuristic (e.g., in Table 2). This reviewer assumes that they refer to Pearson correlation coefficients. If that is the case, were the items not polytomous and ordinal (ordered categorical), in which case polychoric correlations would have been the more accurate coefficient to apply?

5. The authors indicate that, since the variables show a skewed distribution, it became necessary to perform a logarithmic transformation. Could the authors add a graphic plot (perhaps as an appendix) demonstrating that a logarithmic transformation provided better fit to the data than alternatives such as a gamma distribution or polynomial function?

6. In addition to the quantile-quantile plots of the residuals, could the authors also provide findings from statistical tests of normality (e.g. Kolmogorov-Smirnov (K-S) test, Shapiro-Wilk test, Anderson-Darling test, Cramer von Mises test) as further evidence of non-normality?

7. Please correct the grammatical error in the final sentence of page 15, lines 367-369.

8. If the variables are ordinal, Cronbach's coefficient alpha may not be the most appropriate test of internal consistency reliability. In that case, please consider alternatives such as the Ordinal coefficient alpha and McDonald's omega coefficient of composite reliability.

6. PLOS authors have the option to publish the peer review history of their article (what does this mean?). If published, this will include your full peer review and any attached files.

Reviewer #1: **Yes: **Anthony C. Waddimba, MD, DSc.

---

## [Author Response · Author response to Decision Letter 0]

28 Jan 2023

We thank the reviewer for his/her overall positive evaluation of our manuscript, his/her thorough comments and constructive feedback. We have worked through the comments on a point-to-point basis; our answers are provided below in italics.

Reviewer

1. In the introduction section you have referenced various existing item response theory (IRT)-based algorithms, that are reported in the extant literature, for constructing parallel test forms. Given the well known advantages of IRT over classical test theory (CTT) methods, how then do you justify a heuristic that is predominantly founded on CTT? Please enumerate the advantages that your CTT-based heuristic is expected to have over the published IRT-based algorithms? The authors need to build a stronger justification for a CTT-based heuristic.

Thank you very much for this helpful feedback. We realized that we did not highlight the advantages of our heuristic and unintentionally distracted from it by discussing additional measures of the CTT. These were intended merely as a means of demonstrating that the resulting test versions could also meet requirements of the CTT. Of more specific importance to our heuristic is the testing of multivariate equivalence, which we performed by means of a MANOVA and a multivariate Kruskall-Wallis test. 

Following the Reviewer’s suggestion, in the revised manuscript we shortly discuss the advantages of the IRT and elaborate more clearly on the expected advantages of the proposed heuristic, specifically:

Page 1, Introduction:

“Thereby, classical Test Theory (CTT) assumes that each item is equally difficult [1]. While CTT attempts to estimate the true score of the characteristic to be measured based on the responses in several items and focuses on the accuracy of a given measurement as well as on the magnitude of the measurement error, it can only address one variable at a time [2]. (…) Unlike CTT, item response theory (IRT) does not assume that each item is equally difficult [1]. The difficulty of each item is treated as an information that is incorporated into the item characteristic curves (ICC). Thereby, the ICC represents the difficulty of the item by the probability curve of answering the item correctly as a function of the subject's underlying trait. As opposed to CTT, which focuses on the test, IRT focuses on the item. To date, several automated algorithms for constructing parallel test forms that make use of the item information function from IRT have been suggested [4–9]. IRT assumes manifest variables (i.e., the response behavior to test items) and a latent variable (i.e., an underlying characteristic of the subjects). Despite its clear advantages (e.g., items with different difficulty levels and sample independence of test characteristics), IRT approaches usually assume only one latent variable, which is reflected in the correlation between the manifest variables [10–13]. Importantly, both CTT and IRT usually consider only one variable, respectively one latent variable at a time. In practice, however, it can be of interest to describe the items with respect to several variables, to derive more than one latent variable, and to divide these items among parallel test versions such that they are comparable with respect to all variables. (…).”

Page 10, Discussion:

“The core of our heuristic is based on the reduction of multivariate items to two dimensions, which is neither a CTT nor an IRT approach, but could be considered related to IRT. Similar to IRT, we assume that items vary in difficulty, which in our case is represented by the different values of the items on the first principal component (Fig 2).

Our items and variables, on which these principal components are based, were generated, and collected in the past. Among them, Response Time and Accuracy would be the variables most comparable to typical item response variables. These two, along with Image Agreement, form the first principal component, which accounts for 40.79% of the variance in the data. This noteworthy relationship might have escaped our attention, had we used the IRT approach. 

Given that unidimensionality is considered a prerequisite for IRT and bidimensionality is a prerequisite for our dimension-reduction heuristic, it seems a legitimate question how high the proportion of explained variance should be in order to be considered an indicator of unidimensionality or bidimensionality.

Hattie [36] refers to authors who propagate 20% or 40% for unidimensionality. However, in his conclusion, he suggests that it may be unrealistic to search for indications of unidimensionality, and that the test score is basically a weighted composite of all the underlying variables. We share and address this idea by suggesting the use of multivariate items and dimension-reduction procedures.

A common criterion for dimension-reduction methods is to retain as many components until about 70-90% of the variance is explained [21–23]. In our heuristic, by relaying on two principal components, we achieve an explanation of the variance of 65.36%. Whereas the first principal component (composed of Response Time, Accuracy, and Image Agreement) accounts for 40.79% of the variance in the data, the second principal component (composed of Image Complexity and Object Familiarity) accounts for 24.57% of the variance in the data. Even if the latter has no direct influence on Response Time or Accuracy, it may still influence the subjects’ responses, e.g., in the form of faster fatigue over the duration of a test.

In summary, while IRT usually assumes one latent variable, we assume two principal components from a variety of variables that ideally cover a large portion of the variance in the data. This approach helps to represent items on a two-dimensional graph, in which similarity of items is represented by spatial proximity.”

Page 11, Discussion:

“In order to confirm that the resulting test versions met multivariate as well as CTT criteria (i.e., multivariate equivalence, parallelism, equivalence of means, reliability, and internal consistency), we performed both multivariate procedures (i.e., MANOVA and multivariate Kruskal-Wallis) as well as univariate procedures (i.e., Bradley-Blackwood test, TOST, ANOVA, Pearson's product moment correlations, and Cronbach's alpha), most of which are known from CTT. The usual iterative process of the item selection of the CTT could be omitted. (…) Applying a MANOVA as well as a multivariate Kruskal-Wallis test, which better fits our data, we simultaneously examined all variables, and demonstrated that the ABCD test versions did not significantly differ with respect to any of the variables considered. (…) The distinctive feature of the proposed heuristic is that it allows for the generation of multiple test versions while taking several variables into account that have more than one underlying latent variable. Yet, for reasons of practicability, we did not assess all variables with all methods. In our verification approach, except for MANOVA and multivariate Kruskal-Wallis, we focused on the variable response time.”

2. Before applying the PCA, why did the authors not begin by deriving the Kaiser–Meyer–Olkin (KMO) test to confirm adequacy of the data as a source of factor-analytic correlation matrices?

We thank the Reviewer for raising this point. Indeed, the overall Kaiser-Meyer-Olkin (KMO) value of the data is 0.608, deeming it sufficient for factor analysis (FA). Given that PCA is a method of transforming original correlated variables into new, uncorrelated principal components, the correlation between the variables involved is usually considered the most important prerequisite for PCA. Similarly, FA is primarily used to find latent variables, with the KMO value indicating whether the partial correlations are sufficiently small for FA. Moreover, as Dziuban & Shirkey (1974) pointed out, the KMO value improves with a higher number of variables and subjects, a higher level of correlations, and a lower number of factors. We agree with the Reviewer that these prerequisites are very useful for our procedure, and we report the KMO in the revised manuscript. 

Page 4, Results:

“The overall Kaiser-Meyer-Olkin value was sufficient to perform factor analytic procedures (KMO = 0.608).”

3. In what ways is PCA (even robust PCA) superior to other CTT methods such as Horn's parallel analysis or Velicer's minimum average partial (MAP) test, as a foundation for the proposed heuristic?

Both Horn's parallel analysis or Velicer's minimum average partial (MAP) test appear very useful in finding the ideal number of components or factors. However, in our heuristic we had to limit ourselves to two components in order to depict them in a graphical form. Any other dimension-reduction method that allows a two-dimensional graphical representation would also be suitable and is expected to deliver similarly results. We chose PCA in an exemplary fashion due to its widespread use and acceptance in the social sciences, along with its ease of visualization and its suitability for our numerical data. We summarize this on page 4 of the revised manuscript: 

“The main advantages of PCA are its suitability for numerical data, its lack of requirements with respect to distributional assumptions, its suitability for highly correlated variables, along with its usefulness even in the case of relatively large number of variables with respect to observations [21–23].”

Further, we agree with the Reviewer that a comparison of different methods would be of relevance and could be addressed in follow-up studies (see pages 2 and 12 of the revised manuscript):

“In their work, Guasch et al. [14] used k-means clustering, yet other dimension-reduction methods are also applicable, e.g., Multidimensional Scaling (MDS) in the case of mixed data [19], or newer methods for numerical data such as e.g., Stochastic Neighbor Embedding (t-SNE) [15]. Given its wide and common use in psychological research, in the current work we used Principal Component Analysis (PCA) as a dimension-reduction method.”

“Simulation studies and replications using different kinds of data and dimension-reduction methods, such as Multidimensional Scaling (MDS) in the case of mixed data 19], or Stochastic Neighbor Embedding (t-SNE) [15] in the case of numerical data, would further help to prove the generalizability of the proposed heuristic.”

4. The authors do not always specify which correlation coefficients they utilized in their practical application of the proposed heuristic (e.g., in Table 2). This reviewer assumes that they refer to Pearson correlation coefficients. If that is the case, were the items not polytomous and ordinal (ordered categorical), in which case polychoric correlations would have been the more accurate coefficient to apply?

This is an important point and a very helpful advice, thank you. Indeed, we used Pearson's correlation coefficients. We did so under the assumption, which is common in psychometric research, that a Likert scale with at least 5 points may be considered as interval-scaled and treated as continuous in the analysis. However, prompted by the Reviewer’s comment, we realized that this is justifiably controversial (Allen & Seaman, 2007; Jamieson, 2004; Joshi et al., 2015; Leung, 2011; Sullivan & Artino, 2013; Wu & Leung, 2017). As such, we have computed the polychoric correlations, which yielded the same results as Pearson's correlation coefficients (see the revised Table 2 on page 5). 

Table 2: Pairwise comparisons between all variables. 

 ObCl NaAg ImAg ImCo ObFa ACC RT

ObCl – 

NaAg -.26 ** – 

ImAg .22 * .04 – 

ImCo .07 .10 .01 – 

ObFa .41 *** -.15 .17 .54 *** – 

ACC .18 .00 .71 *** .07 .27 ** – 

RT -.06 .36 *** .59 *** .12 .08 .68 *** –

Note. Correlations indicate that PCA is applicable. ObCl = object class, NaAg = naming agreement, ImAg = image agreement, ImCo = image complexity, ACC = accuracy, RT = response time. Significance levels: *** < 0.001, ** < 0.01, * < 0.05. Test statistics based on Pearson's product moment correlations and polychoric correlation coefficients.

5. The authors indicate that, since the variables show a skewed distribution, it became necessary to perform a logarithmic transformation. Could the authors add a graphic plot (perhaps as an appendix) demonstrating that a logarithmic transformation provided better fit to the data than alternatives such as a gamma distribution or polynomial function?

Following the Reviewer’s suggestion, in the revised manuscript, next to the quantile-quantile plot of the Response Time and its logarithmic-transformation, we have added a box-cox-transformation plot, which yielded very similar results to log-transformation. This is indicated in the revised Figure 4 and on page 8 of the revised manuscript: 

Fig 4: Quantile-Quantile Plot of response time (left), logistic-transformed response time (middle), and box-cox-transformed response time (right).

“We found logarithmic and box-cox-transformation to deliver a very similar result.”

6. In addition to the quantile-quantile plots of the residuals, could the authors also provide findings from statistical tests of normality (e.g. Kolmogorov-Smirnov (K-S) test, Shapiro-Wilk test, Anderson-Darling test, Cramer von Mises test) as further evidence of non-normality?

This is an important note, thank you. We have implemented this suggestion in the revised manuscript.

Page 3:

“To assess whether all variables were equally well addressed by the PCA, perform a multivariate comparison of the final parallel test versions, such as analysis of variance (MANOVA) or multivariate Kruskal-Wallis.“

Page 6:

“Although according to the visual inspection of the quantile-quantile plot, the log- as well as the box-cox-transformed variable response time seemed normally distributed, the normal distribution assumption had to be rejected with the Kolmogorov-Smirnov (K-S) test (p-value < .001).”

Page 7:

“Since the proposed method to generate parallel tests takes multiple variables into account, we further propose a parametric multivariate analysis of variance (MANOVA) or a non-parametric multivariate Kruskal-Wallis test as a mean to compare test versions. (…) 

For a multivariate comparison of the test versions ABCD, both a MANOVA and a multivariate Kruskal-Wallis (MKW) test were performed on the items’ data, since the normality assumption was rejected by the K-S test. (…) If a given response variable had been significant in the MANOVA or MKW test, it would have meant that at least one test version was different from the others concerning this particular variable. (…) As shown in Table 3, the test versions did not significantly differ on any of the considered variables (p = 1.00 for ABCD for all variables), with naming agreement showing the lowest p-value (MANOVA: p = 0.75, MKW: p = 0.54).”

Table 3: Multivariate comparison of the parallel test versions ABCD. 

 ObCl NaAg ImAg ImCo ObFa ACC RT All variables

MANOVA 1.00 0.75 0.99 0.84 0.80 0.95 0.99 1.00

MKW 1.00 0.54 0.99 0.97 0.87 0.86 0.95 1.00

Note. Abbreviations: ObCl = object class, NaAg = naming agreement, ImAg = image agreement, ImCo = image complexity, ACC = accuracy, RT = response time, MKW = Multivariate Kruskal-Wallis test. Numbers are indicating p-values.

7. Please correct the grammatical error in the final sentence of page 15, lines 367-369.

We thank the Reviewer for spotting this grammatical infelicity. We have replaced ‘hold’ with ‘held’ in the revised manuscript (now page 9, line 298). 

8. If the variables are ordinal, Cronbach's coefficient alpha may not be the most appropriate test of internal consistency reliability. In that case, please consider alternatives such as the Ordinal coefficient alpha and McDonald's omega coefficient of composite reliability.

We agree with the Reviewer that the Likert-scaled data should be considered and treated ordinally. We have therefore included a non-parametric multivariate Kruskall-Wallis test in the revised manuscript, since it includes the likert-scaled variables (see also our response to point 6). However, we have only analyzed Cronbach's alpha for the response time, for which we assumed interval scaling. 

Furthermore, in this study, the calculation of Cronbach’s alpha serves solely confirmatory purposes and aims at demonstrating that the test versions resulting from the heuristic can also meet the requirements of CTT. In the case of internal consistency, however, we find it questionable whether it needs to be met at all. We elaborate on this on page 11 of the revised Discussion:

“Given parallelism, these can be considered reliability and internal consistency measures. However, Cronbach's alpha is a typical CTT measure that reflects the degree to which items within a test version are similar with respect to that dimension. This is not central to our heuristic; on the contrary, we want to allow for item difficulty to vary within a test version, while not leading to different difficulty distributions across test versions. The fact that we nevertheless obtained good results in terms of internal consistency is probably due to the homogeneity of the items in terms of response time and to the fact and that the items were mostly selected from the center of the biplot.”  

Journal Requirements

Thank you. We deleted the key words from our revised manuscript, as they seem not to be included in the style templates. 

2. Thank you for stating in your Funding Statement: “Nicole Göbel had financial support by the SNF Grant No: 175615.” Please provide an amended statement that declares *all* the funding or sources of support (whether external or internal to your organization) received during this study, as detailed online in our guide for authors at http://journals.plos.org/plosone/s/submit-now. Please also include the statement “There was no additional external funding received for this study.” in your updated Funding Statement. Please include your amended Funding Statement within your cover letter. We will change the online submission form on your behalf.

Thank you very much for updating our funding statement. It has been deleted from the Acknowledgements and added to the cover letter as follows:

“The study was supported by the Swiss National Science Foundation Grant no: 175615. There was no additional external funding received for this study.”

We have transferred the revised manuscript into the LaTeX template. Unfortunately, since we lack experience with LaTeX, next to the revised LaTeX file (i.e. Revised Manuscript.tex and Revised Manuscript.pdf), we also submit a word file where the changes are indicated (i.e. Revised Manuscript with Track Changes.docx). 

4. Thank you for stating the following in the Acknowledgments Section of your manuscript: “We would like to thank Dr. Dorothea Weniger for her expert advice in 440 linguistics. We would also like to thank Dr. Lea Jost for valuable discussions on 441 methodology. The study was supported by the Swiss National Science Foundation 442 Grant no: 175615” We note that you have provided additional information within the Acknowledgements Section that is not currently declared in your Funding Statement. Please note that funding information should not appear in the Acknowledgments section or other areas of your manuscript. We will only publish funding information present in the Funding Statement section of the online submission form. Please remove any funding-related text from the manuscript and let us know how you would like to update your Funding Statement. Currently, your Funding Statement reads as follows: “Nicole Göbel had financial support by the SNF Grant No: 175615.” Please include your amended statements within your cover letter; we will change the online submission form on your behalf.

Thank you very much for updating our funding statement. It has been deleted from the Acknowledgements and added to the cover letter. 

5. We note that you have referenced (Berger E-M.et al. [21]) which has currently not yet been accepted for publication. Please re We note that you have referenced (ie. Bewick et al. [5]) which has currently not yet been accepted for publication. Please remove this from your References and amend this to state in the body of your manuscript: “Berger E-M et al. [Unpublished]”) as detailed online in our guide for authors http://journals.plos.org/plosone/s/submission-guidelines#loc-reference-style move this from your References and amend this to state in the body of your manuscript: (ie “Bewick et al. [Unpublished]”) as detailed online in our guide for authors http://journals.plos.org/plosone/s/submission-guidelines#loc-reference-style

Thank you, we have implemented the suggested changes on page 8 and in the References.

Journal Requirements

1. Thank you for including your ethics statement on the online submission form:

"The studies involving human participants were reviewed and approved by the Ethics Committees of the Cantons of Luzern and Bern, Switzerland (EKNZ 2015-256; KEK BE 151/15). The patients/participants provided their written informed consent to participate in this study."

To help ensure that the wording of your manuscript is suitable for publication, would you please also add this statement at the beginning of the Methods section of your manuscript file.

We have added the ethics statement at the beginning of the methods with minor additions:

"The studies involving human participants were reviewed and approved by the Ethics Committees of the Cantons of Luzern and Bern, Switzerland (EKNZ 2015-256; KEK BE 151/15). The participants provided their written informed consent to participate. The data were analyzed anonymously for this study."

2. We note the OSF link you have provided is view-only. Please note we require all data to be publicly available without login credential requirements. As such, please unlock your OSF data and provide us with the updated DOI/URL.

Thank you for pointing this out. We have made the material publicly available on https://osf.io/3a4c5/ with DOI 10.17605/OSF.IO/3A4C5.

---

## [Decision Letter · Decision Letter 1]

10 Apr 2023

An item sorting heuristic to derive equivalent parallel test versions from multivariate items

PONE-D-22-04765R1

Dear Dr. Göbel,

We’re pleased to inform you that your manuscript has been judged scientifically suitable for publication and will be formally accepted for publication once it meets all outstanding technical requirements.

Kind regards,

Alessandro Barbiero, Ph.D. in Statistics

Academic Editor

PLOS ONE

Additional Editor Comments (optional):

Reviewers' comments:

Reviewer's Responses to Questions

**Comments to the Author**

1. If the authors have adequately addressed your comments raised in a previous round of review and you feel that this manuscript is now acceptable for publication, you may indicate that here to bypass the “Comments to the Author” section, enter your conflict of interest statement in the “Confidential to Editor” section, and submit your "Accept" recommendation.

Reviewer #1: All comments have been addressed

2. Is the manuscript technically sound, and do the data support the conclusions?

Reviewer #1: Yes

3. Has the statistical analysis been performed appropriately and rigorously? 

Reviewer #1: Yes

4. Have the authors made all data underlying the findings in their manuscript fully available?

Reviewer #1: Yes

5. Is the manuscript presented in an intelligible fashion and written in standard English?

Reviewer #1: Yes

6. Review Comments to the Author

Reviewer #1: Thank you for responding to the reviewers' comments. Good luck with your ongoing research. I look forward to seeing more of your work in future publications.

7. PLOS authors have the option to publish the peer review history of their article (what does this mean?). If published, this will include your full peer review and any attached files.

Reviewer #1: **Yes: **Anthony C. Waddimba, MD, DSc.

---

## [Editor Report · Acceptance letter]

14 Apr 2023

PONE-D-22-04765R1 

An item sorting heuristic to derive equivalent parallel test versions from multivariate items 

Dear Dr. Göbel:

I'm pleased to inform you that your manuscript has been deemed suitable for publication in PLOS ONE. Congratulations! Your manuscript is now with our production department. 

Kind regards, 

on behalf of

Dr. Alessandro Barbiero 

Academic Editor

PLOS ONE